# Electroplastic Effects on the Mechanical Responses and Deformation Mechanisms of AZ31 Mg Foils

**DOI:** 10.3390/ma15041339

**Published:** 2022-02-11

**Authors:** Shuai Xu, Xinwei Xiao, Haiming Zhang, Zhenshan Cui

**Affiliations:** School of Materials Science and Engineering, Shanghai Jiao Tong University, 800 Dongchuan Road, Shanghai 200240, China; xushuai@sjtu.edu.cn (S.X.); realxiaoxw@gmail.com (X.X.); cuizs@sjtu.edu.cn (Z.C.)

**Keywords:** electroplastic effect, size effect, deformation twinning, fracture characteristics, magnesium alloys

## Abstract

Electrical-assisted (EA) forming technology is a promising technology to improve the formability of hard-deformable materials, such as Mg alloys. Herein, EA micro tensile tests and various microstructure characterizations were conducted to study the electroplastic effect (EPE) and size effect on the mechanical responses, deformation mechanisms, and fracture characteristics of AZ31 Mg foils. With the assistance of electric currents, the ductility of the foils was significantly improved, the size effects caused by grain size and sample thickness were weakened, and the sigmoidal shape of the flow stress curves during the early deformation stage became less obvious. The EBSD characterization results showed that the shape change of the flow stress curves was due to the EPE suppressing the activation of extension twinning at the early deformation stage, especially for the coarse grain samples. The suppression of extension twinning resulted in a quick increase in flow stress due to the dislocation-dominant work hardening, and the increased flow stress eventually promoted extensive deformation twins at large deformation. Thus, as the sample strained to 10% tensile deformation, the EA-tested samples showed a larger volume fraction of deformation twins than the non-EA samples. The reference orientation deviation analysis verified that the deformation twins in the EA samples were formed in the large deformation stage. Combined with the fractography, the EPE also improved the ductility by suppressing the expansion of cleavage surfaces.

## 1. Introduction

Magnesium (Mg) alloys are of growing interest in micro-parts for medical purposes and microelectromechanical systems (MEMS) for their high specific strength and excellent biodegradability [1]. However, for the low-symmetric hexagonal close-packed (HCP) crystal structure with limited slip systems at room temperature, they exhibit poor formability that restricts their wide applications [2]. Compared with the cubic metals, e.g., aluminum, steel, and copper, Mg alloys show more complex plasticity behaviors generally involving two types of deformation modes, i.e., dislocation slip and deformation twinning. At room temperature, basal 〈a〉 slip and prismatic 〈a〉 slip are the main deformation modes which, however, can only coordinate the deformation along the *a*-axis. To coordinate the deformation in the *c*-axis, $\left\{10\bar{1}2\right\}\langle 10\bar{1}1\rangle$ extension twinning is commonly activated in Mg alloys, and it can also reorient the *c*-axis of parent grains around 86.3° [3]. The shear strain due to extension twinning is not large enough; it is only about 0.13 [4]. Moreover, in the processes of part miniaturization, the size effects caused by grain size and sample size further complicate the deformation mechanisms of Mg alloys, and the formability will be worse [5,6,7].

Electrical-assisted (EA) plasticity is regarded as an effective way to improve the formability of hard-deformable metals [8,9,10,11,12]. For example, Stolyarov et al. [8] reported that the yield and ultimate stresses of TiNi alloys were improved during the electroplastic rolling process. Mai et al. [9] developed an electrical-assisted embossing process to fabricate micro-channels on metal workpieces of 316L stainless steel. With the assistance of high-density electric currents, the flow stress of the material was decreased, and the depth of the fabricated channel feature was increased during the embossing process. Xie et al. [10] reported that the springback of AZ31B Mg alloy sheets during V-bending tests can be reduced by the assistance of electric currents. Based on the springback angle prediction model they proposed, the springback reduction was attributed to the detwinning that occurred in the EA stress relaxation after V-bending processes. Li et al. [11] improved the formability of AZ31 Mg alloys during the gas bulging process with the assistance of pulse currents; they found that pulse current could prevent the dislocation tangle and restrain the cavity growth during the bulging process. Sánchez et al. [12] improved the formability of 308L stainless steel up to 11.9% and the relative energy efficiency up to 7.6% by inducing electro-pulses into the wire drawing process. Consequently, EA metal-forming operations are of growing interest as an important solution to form hard-deformable metals and alloys [13,14,15], and many works are dedicated to study the specific roles of the electro-plasticity effect (EPE) in the deformation and mechanical response of materials [16].

It is commonly accepted that the mechanisms behind the EPE on the deformation and mechanical properties of metals mainly includes the Joule heating effect and the athermal current-induced effect. Troitskik et al. [17] firstly attributed the change in mechanical behaviors and microstructure evolution to the non-thermal effect of electric currents under EA conditions; they suggested that the electrons could transfer the energy to microstructural defects such as dislocations and improve the localized diffusion rate and the localized heating, thus improving the formability. Perkins et al. [18] later separated Joule heat from the EA deformation via traditional thermal experiments and found that Joule heat is not the sole determinant of the EPE. Conrad et al. [19] and Kim et al. [20] proved that the interaction of currents with atoms and dislocations could improve the plasticity during the EA deformation. Wang et al. [16] found that the softening effect due to electric currents was not exclusive to the bulk thermal softening during the EA micro-forming of AZ31 Mg alloys, and they developed a composite material model considering the localized Joule heating at grain boundaries. Magargee et al. [21] established modified Hollomon and Johnson–Cook models of flow stress to predict the softening of materials assisted by the direct current and the increase in temperature caused by Joule heating during the EA tension tests. Although electroplasticity has been well recognized and widely applied in modern manufacturing industries, the underlying physics governing the EPE on the deformation mechanism and microstructure evolution of materials is not yet well understood and is controversial to some extent, especially for the HCP metals such as Mg-based alloys, which have complicated plasticity behaviors generally involving the concurrence of dislocation slip and deformation twinning on various crystallographic slip/twin systems [22,23].

The plastic deformation and formability of Mg alloys are rather sensitive to the intrinsic microstructure feature and external boundary constraints [5]. For instance, deformation twinning is strongly affected by grain size and is more easily activated as grain size increases [5,7]. Prasad et al. [24] reported that a remarkable specimen size effect occurred in Mg single crystals subjected to uniaxial compression tests, and that the resistance of twinning of the micropillar sample is several orders of magnitude larger than that of the macroscopic sample. Nevertheless, there are few studies focusing on the EPE coupled with the micro-/meso-scale plastic deformation of Mg foils. This work aims to study the EPE on the mechanical responses and deformation mechanisms of commercial AZ31 Mg foils. Both non-EA and EA micro-tensile tests were performed on Mg foils with different grain sizes and sample thicknesses; electron backscatter diffraction (EBSD) and scanning electron microscope (SEM) characterizations were conducted to capture the microstructure evolutions and the fracture behaviors of the deformed samples.

## 2. Sample Preparation and Experimental Details

The as-received material was a commercial extruded AZ31 Mg alloy. It was refined by equal channel angular pressing (ECAP) operations and annealed under different heat treatments to obtain samples with various grain sizes. The grain size (*d*) of the samples annealed at 450 °C for 10 h was about 21 μm (coarse grain) and the grain size of the samples annealed at 250 °C for 2 h was about 9 μm (fine grain). The annealed samples were cut into miniature plate dog bone-shaped specimens with thicknesses *t* = 0.5 mm and 1.0 mm, as shown in Figure 1a. More detailed information about the sample preparation can be found in our previous work [5,22].

The EA micro-tensile tests were performed on a micro-tester at a constant crosshead speed of 0.3 mm/min and assisted with direct current (DC) pulse currents of varying intensities, i.e., J = 10, 30, 40, and 50 A/mm^2^; the EA tensile tester is shown in Figure 1b,c. The tensile direction is also the extrusion direction (ED) of the ECAP operations. To exclude the influence of Joule heat, an air blower blowing liquid nitrogen was used to keep the sample’s temperature around the ambient temperature of 25 °C. The non-EA samples and the samples assisted with an electric current density of 40 and 50 A/mm^2^ were elongated to a global strain of $\varepsilon_{\mathrm{ED}}=$ 10%, and their microstructure evolutions were measured via EBSD characterization. The fracture characteristics of the samples under various deformation conditions were characterized via second electron (SE). The EBSD and SE were characterized using a Vega thermal field emission SEM. The samples for EBSD characterizations were first ground with the grit SiC papers with different grades and then electro-polished at a 15 V voltage in 3% perchlorate alcohol solution chilled to −40 °C.

## 3. Results and Discussions

### 3.1. Initial Microstructure

Figure 2 shows the initial microstructures and the grain size distribution of the samples with different grain sizes in terms of OIMs with ED inverse pole figure (IPF) coloring. The average grain size was about 21 μm for the coarse grain samples and 9 μm for the fine grain samples. Both the coarse grain samples and fine grain samples had undergone enough recrystallization and exhibited twin-free equiaxial grain structures. The ED and TD IPFs show that the annealed samples had a similar fiber texture and that the (0001) pole was about a 20° deviation from the ED. Since the ED is also the tensile direction, such a fiber texture in the Mg alloy should facilitate the activation of extension twinning at the early stage of the tensile deformation.

### 3.2. Size Effect and EPE on the Flow Stress Curves of the Material

Figure 3 presents the true stress–strain curves of both the non-EA and EA micro-tensile samples. To ensure the repeatability of the experiments, each condition was repeated at least three times. As shown in Figure 3, the true stress–strain curves of the non-EA tensile samples all exhibit a concave downward shape during the early deformation stage, indicating the activation of plentiful deformation twins. Meanwhile, the non-EA samples show the fine grain strengthening effect of grain size and the “smaller is weaker” trend of sample thicknesses, which is consistent with our previous work [5].

However, the size effects caused by grain size and sample thickness were weakened with the assistance of electric currents. The sigmoidal shape feature of the flow stress curves during the early deformation stage was attenuated with the increase in current intensity. Therefore, it can be inferred that the electric current suppressed deformation twinning at the early deformation stage. Since the activation and expansion of extension twinning can release the stress concentration during the plastic deformation, once the extension twinning is suppressed, the work hardening resulting from the dislocation slip will be enhanced. Furthermore, one can see that the yield strength increased with the assistance of electric currents, especially for the coarse grain samples (in Figure 3a,b); the yield strength even increased from ~70 MPa to ~110 MPa. Although the EPE improved the yield strength, the samples with the same thickness and grain size were basically the same under different currents, i.e., the degree of improvement in yield strength is not sensitive to the current intensity.

In contrast, the ultimate tensile strength (UTS) is rather sensitive to the current intensity. Though there is a common rule that the UTS decreases with the increase in current intensity in this study, the EPE on the UTS of the samples with different thicknesses and grain sizes was quite different. For the coarse grain samples with *t =* 0.5 mm (Figure 3a), when the current intensity was small, e.g., J = 10 and 30 A/mm^2^, the UTS of the EA samples was higher than that of the non-EA samples. For the coarse grain specimens with *t =* 1.0 mm (Figure 3b), all the EA samples exhibited a higher UTS than the non-EA samples. For the coarse grain samples, the effect of sample thickness on the UTS could be attributed to the different mechanical properties between the surface grains and the interior grains [25]. With the decrease in sample thickness, the volume fraction of surface grains increased. The surface grains may be more sensitive to the EPE, and in the EA samples with *t* = 0.5 mm, the surface layer can weaken the UTS more obviously with the increase in current intensity. The same phenomenon that the EPE intensified the softening effect of the surface layer on the global strength is found in the fine grain samples, as shown in Figure 3c,d. By comparing the flow stress curves of the fine grain and coarse grain samples under the EA tensile tests, one can see that the “fine grain strengthening effect” disappeared in the EA conditions. The EA fine grain samples exhibited a much lower UTS than that the coarse grain samples at higher current densities (i.e., ≥40 A/mm^2^). Grain boundary (GB), as a crystal defect, exhibits a higher electrical resistance than the interior grains during the EA deformation. The volume fraction of GB increases with the decrease in grain size, and more localized Joule heating will be generated at the GBs during the EA deformation [16]. Thus, a more significant softening effect occurred in the fine grain samples. Moreover, the softening effects from the Joule heating generated at GBs could also improve the ductility; as shown in Figure 3a–d, the elongation of the EA-tested samples was improved with the increase in current intensity. Especially for the fine grain samples with *t* = 0.5 mm, the elongation was increased by approximately 60% when the assisted electric current density was 50 A/mm^2^.

### 3.3. Deformation Mechanisms Associated with Size Effects and EPE

To capture the microstructure evolution, the deformed samples after elongation of 10% were examined on the ED-TD surface via EBSD characterization. The obtained orientation imaging microscopies (OIMs) and the band contrast (BC) maps with GBs and twinning boundaries (TBs) are shown in Figure 4, including the non-EA samples and EA (the electric current density is 40 A/mm^2^) samples with different grain sizes and thicknesses. The black lines represent high-angle GBs (with a misorientation angle above 15°) and the white lines represent TBs. According to the reorientation analysis, all the TBs are the boundaries of extension twins, as the misorientation angle between the twin–matrix interface is approximately 86.3 ± 5°. The initial texture features (as shown in Figure 2) essentially disappeared; that is, all the samples underwent a significant texture evolution after 10% tensile deformation. As shown in Figure 4a,d, with the increase in sample thickness or the decrease in grain size, the activation and expansion of extension twins were suppressed significantly. For the coarse grain samples, as shown in Figure 4a–c, various twins grew and merged and occupied a large volume fraction in the parent grains. By contrast, deformation twins in the fine grain sample were evidently suppressed and numerous twin embryos appeared at the GBs, as shown in Figure 4. All the characterized twins had orientation with the tensile direction close to the $\left[\bar{1}2\bar{1}0\right]$ crystallographic axis; this is the footprint of the reorientation of the extrusion-textured parent grains due to extension twinning.

Comparing the EA coarse grain sample (Figure 4b) with the non-EA coarse grain sample (Figure 4a) and the EA fine grain sample (Figure 4e) with the non-EA fine grain sample (Figure 4f), one can see that the EPE slightly promoted the extension twinning for the samples after a large deformation. The volume fraction of TBs increased from ~43.6% to ~47.4% for the coarse grain samples and from ~16.0% to ~18.6% for the fine grain sample. However, as discussed in Section 3.2, the flow stress curves shown in Figure 3 imply that the EPE significantly suppressed the deformation twinning of the tested sample at the early deformation stage. On the other side, the suppression of deformation twinning resulted in a quicker increase in flow stress, which eventually promoted the activation of deformation twinning for the samples having large deformation. This is consistent with Figure 3 that most EA-tested samples show a larger true stress than the non-EA samples as they strained to 10%. In contrast, the comparisons of Figure 4b versus Figure 4c,e,f verify that the thickness increase largely suppressed the deformation twinning; this phenomenon of geometric constraints was also reported by Prasad et al. [24] and Xu et al. [5]. In short, both grain size and sample thickness significantly affected the activity of deformation twinning. The EPE suppressed the deformation twinning at the early deformation stage through either promoting dislocation slip or the Joule heating. With the proceeding of deformation, the increased flow stress due to the dislocation-dominant work hardening eventually accelerated the activity of deformation twinning.

Based on the EBSD data shown in Figure 4, we present the kernel average misorientation (KAM) maps of the deformed samples in Figure 5. The KAM is a measurement of local grain misorientation associated with local gradient and deformation heterogeneity associated with dislocation slip. The non-EA fine grain thick sample exhibits obviously higher KAM (in Figure 5d) than the non-EA coarse grain thin sample (in Figure 5a). The activation of extension twinning is regarded as an effective way to release the stress concentration during plastic deformation, while the extension twinning was more significantly suppressed in the fine grain thick sample (Figure 4b) than in the coarse grain thin sample (Figure 4a). To coordinate the plastic deformation, more dislocation slip must be activated in the fine grain thick sample, which will result in the high value of KAM. As the tensile deformation was assisted by electric currents, the EA fine grain samples (Figure 5e,f) exhibited less deformation heterogeneity. Meanwhile, the deformation heterogeneity caused by the sample thickness decrease can be eliminated by the EPE, as shown in Figure 5e,f. On the contrary, the EPE showed less influence on the deformation homogeneity of the coarse grain samples, as shown in Figure 5a–c. This is mainly because of the relatively large contribution of deformation twinning in the coarse grain samples. Combined with mechanical behaviors in Figure 3, it is concluded that the relief of deformation heterogeneity with the assistance of electric current improved the ductility of the Mg foils.

To quantitatively describe the influence of size effect and the EPE on the deformation heterogeneity of the Mg foils, the KAM histogram of the deformed samples is shown in Figure 6. It seems that KAM = 1.0° is a critical value at which the volume fraction of the material points with different KAM values obviously changed. In the coarse grain samples, the volume fraction of the material points with KAM less than 1.0° is higher than that in the fine grain samples, and there are more material points with high KAM (≥1.0°) in the fine grain samples. This is because extension twinning was more easily activated in the coarse grain samples, which progressively eliminated the deformation heterogeneity. For the fine grain samples, there are more material points with lower KAM (≤1.0°) in the EA-tested samples. However, the EPE on the deformation heterogeneity is not obvious in the coarse grain samples.

To verify the possible roles of the EPE in affecting dislocation slip and deformation twinning, the grain reference orientation deviation (GROD, the intragranular misorientation angle at each grid relative to the mean orientation of grains) maps of the deformed samples were computed from the EBSD data, and the results are shown in Figure 7. These GROD maps can be interpreted to estimate the shear strain due to dislocation slip in the constrained grains in polycrystals [26,27]. The GROD maps are heterogeneous in all samples. Most deformation twins have a very small GROD angle, which means these twins underwent less dislocation slip after the reorientation. On the contrary, grains without deformation twins have large GROD values; it is anticipated that these grains have crystallographic orientation more favorable to dislocation slip and less to deformation twinning, so they have large shear deformation associated with dislocation slip and larger GROD angles. Furthermore, hot spots (GROD angle close to 15°) primarily occurred in small grains, in which deformation twinning was strongly suppressed due to their very small grain size. A noticeable feature exhibited in Figure 7 is that for the coarse grain samples, the EA samples have more grains with large GROD angles than the non-EA samples; this implies that dislocation slip contributed more plastic deformation in the EA samples. On the other hand, systematic differences among the non-EA and EA fine grain samples are not noticeable.

To reveal the EPE on the deformation modes, we further compared the maps of the crystal directions of the misorientation axes of the non-EA and EA coarse grain samples with thickness t=0.5 mm. The crystal directions of the misorientation axes show the crystal rotation axis during slip deformation. For a single slip, the rotation axis is normal to both the slip plane normal and slip direction of the activated slip system; a $\left[\bar{1}100\right]$ rotation axis (blue in the IPF map) corresponds to 〈a〉 basal slip, a $\left[0001\right]$ rotation axis (red in the IPF map) corresponds to 〈a〉 prismatic slip, and a $\left[\bar{1}2\bar{1}0\right]$ rotation axis (green in IPF map) corresponds to a special multiple-slip case, in which where two adjacent 〈a〉 basal slip systems were activated in similar proportions in the same region [28]. The white regions in Figure 8 are areas with GROD below 2°, which have uncertainty in the misorientation axis measurement. As shown in Figure 8, both maps show large portions of blue or green colors in the un-twinned region, which demonstrates the large contribution of basal slip during the tensile deformation. A noticeable feature in Figure 8 is that there are more deformation twins in the non-EA sample exhibiting an identifiable rotation axis, as highlighted by the ellipses. This indicates that these deformation twins were formed early and contributed the subsequent plastic deformation through dislocation slip. By contrast, there are fewer deformation twins in the EA sample showing an identifiable rotation axis, which means the deformation twins in the EA sample were formed latterly and they underwent less dislocation-dominant deformation as the sample strained to 10% deformation.

### 3.4. Size Effect and EPE on Fracture Behaviors

The fracture characteristics of the non-EA and EA uniaxial tension with different sample thicknesses and different grain sizes were examined and compared, and Figure 9 presents the fractographies of the non-EA tension deformation. For the coarse grain samples, there is a mixed cleavage–ductile fracture surface with dimples, a cleavage surface, and a tearing edge (Figure 9a), and the area of cleavage surface decreased with the increase in sample thickness (Figure 9c). By contrast, the fine grain samples (Figure 9b,d) exhibit typical ductile fracture characteristics that the cleavage surfaces were occupied by dimples and the dimples were distributed more evenly with the increased thickness.

Figure 10 shows the fractography of the EA tensile deformation. Comparing with the fractographies of non-EA tensile samples shown in Figure 9, it can be found that the assistance of electric currents suppressed the expansion of cleavage surfaces. Moreover, the cleavage surfaces can be totally replaced by dimples as the electric current increased, and the dimples were coalesced to form new larger and deeper dimples in both the coarse grain and fine grain samples. During the tensile deformation, the crack source usually tends to appear at the voids, inclusions, and other micro-defects, and these microstructural defects generally have a higher electrical resistance than the matrix. Thus, more localized Joule heat can be generated at the defects, which will result in the localized softening of the material at the junction between the defects and the matrix, and thus, the damage tolerance and ductility can be improved significantly [29].

## 4. Conclusions

The electroplastic effect (EPE) significantly influenced the mechanical responses of AZ31 Mg foils with different grain sizes and sample thicknesses during the EA micro-tension tests. The size effects caused by grain size and sample size were weakened with the assistance of the EPE. The coarse grain samples are more sensitive to electric currents during the tension tests as the yield strength increased, although the ultimate tensile strength decreased. The EBSD characterizations demonstrate that only extension twins were activated during the non-EA and EA tension deformation and the electric current suppressed the activation of extension twins at the early deformation stage, especially for the coarse grain samples. However, as the deformation increased, the flow stress due to the dislocation-dominant work hardening was increased, which eventually resulted in the activation of extensive deformation twins at large deformation. Thus, the EA-tested samples showed a larger volume fraction of deformation twins than the non-EA samples after they were strained to 10% tensile deformation. The reference orientation deviation analysis verified that the deformation twins in the EA samples were formed in the large deformation stage. Meanwhile, the EPE can promote damage tolerance and prevent the expansion of the cleavage surfaces; thus, the formability of the EA tension samples was increased during the EA tension tests. To improve the reliability of EA forming technology in the manufacturing of micro-scaled products, advanced constitutive models considering the size effect and the EPE must be established based on current research, especially the combined effect on coordinating various deformation mechanisms such as dislocation slip and deformation twinning.

## Figures and Tables

**Figure 1 materials-15-01339-f001:**
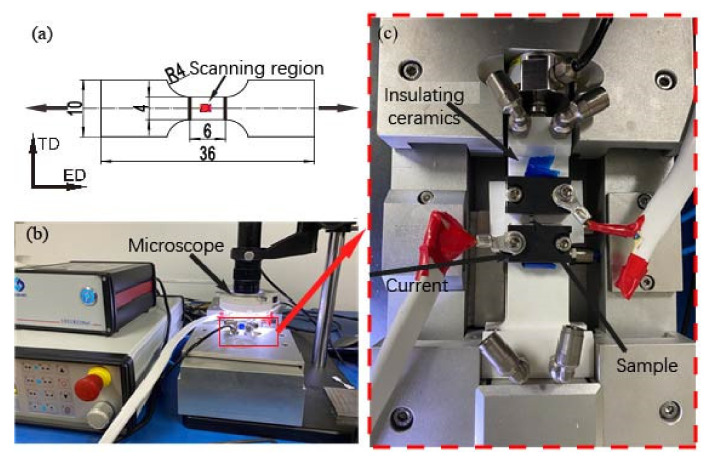
(**a**) Schematic of dog bone-shaped samples for EA uniaxial tensile tests; (**b**,**c**) the EA micro-tension tester.

**Figure 2 materials-15-01339-f002:**
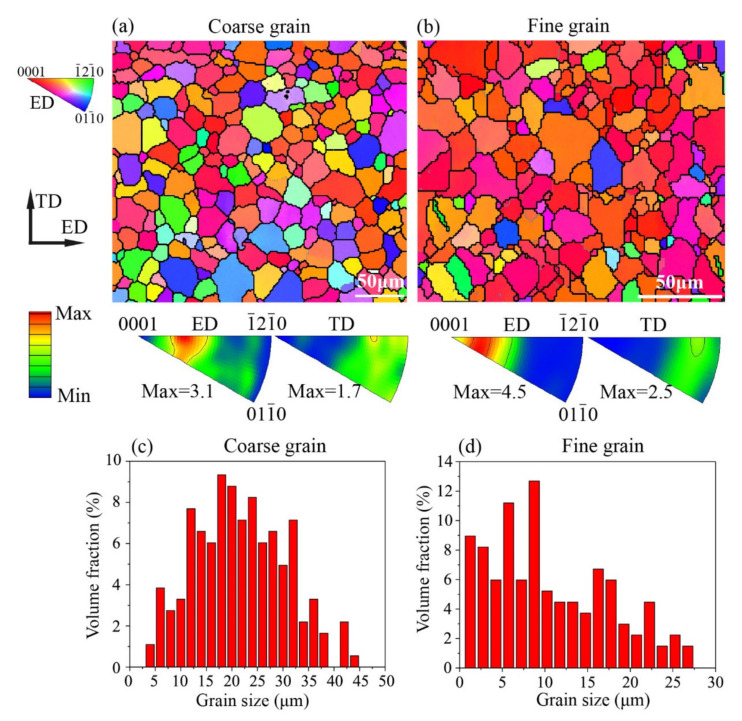
The initial microstructures and the grain size distribution of the samples with different grain sizes in terms of OIMs with the ED inverse pole figure (IPF) coloring and the ED and TD IPFs. (**a**) OIM of coarse grain sample, (**b**) OIM of fine grain sample, (**c**) grain size distribution of the coarse grain sample, and (**d**) grain size distribution of the fine grain sample.

**Figure 3 materials-15-01339-f003:**
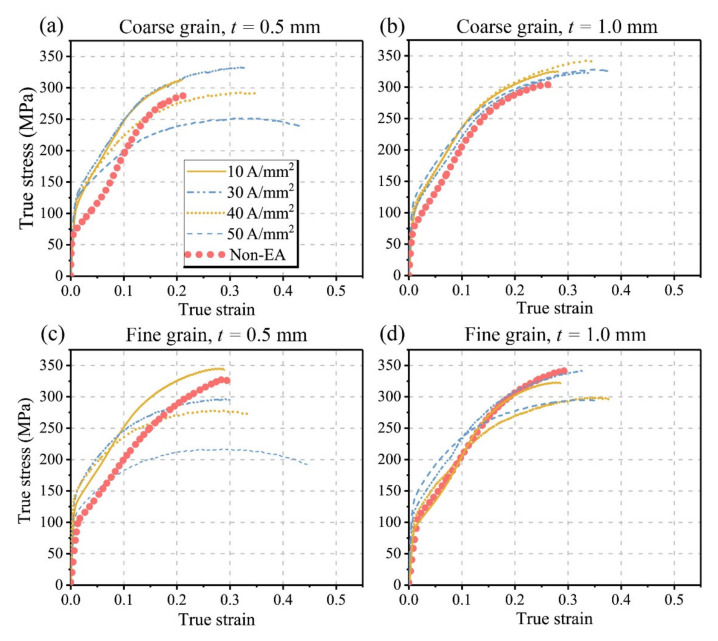
The true stress–strain curves of the EA and non-EA tensile tests of AZ31 foils with different grain sizes and thicknesses. (**a**) Coarse grain, *t* = 0.5 mm; (**b**) coarse grain, *t =* 1.0 mm; (**c**) fine grain, *t* = 0.5 mm; (**d**) fine grain, *t* = 1.0 mm.

**Figure 4 materials-15-01339-f004:**
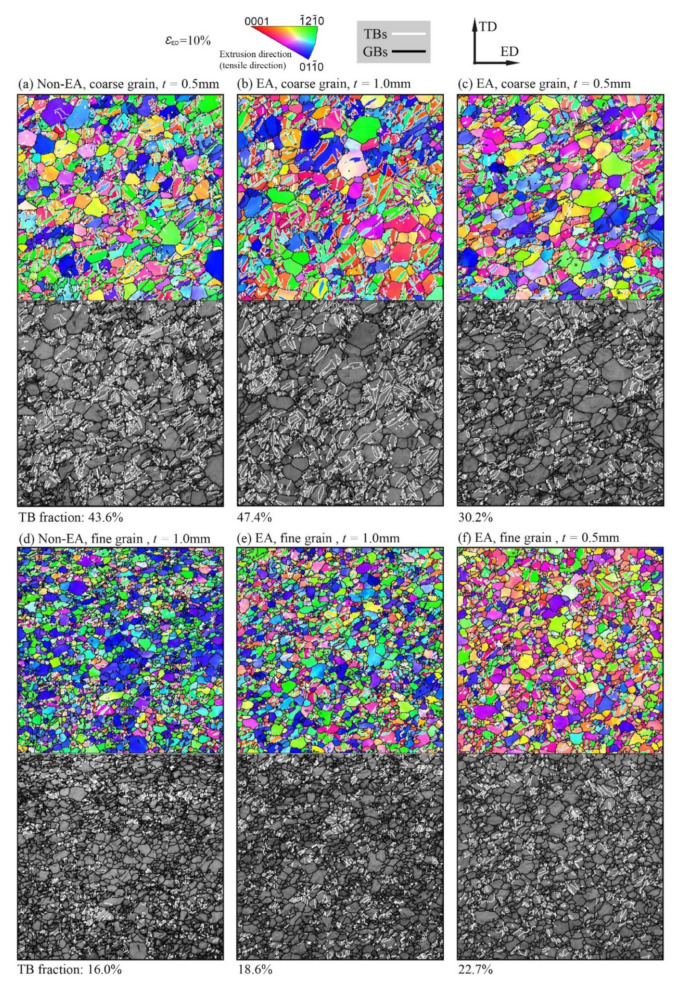
The orientation imaging microscopies (OIMs) and band contrast (BC) maps of the samples strained to 10% tensile deformation under various deformation conditions. (**a**) Non-EA, coarse grain, *t* = 0.5 mm; (**b**) EA, coarse grain, *t* = 0.5 mm; (**c**) EA, coarse grain, *t* = 1.0 mm; (**d**) non-EA, fine grain, *t* = 1.0 mm; (**e**) EA, fine grain, *t* = 1.0 mm; (**f**) EA, fine grain, *t* = 0.5 mm. The black lines represent grain boundaries, and the white lines represent twinning boundaries (TBs).

**Figure 5 materials-15-01339-f005:**
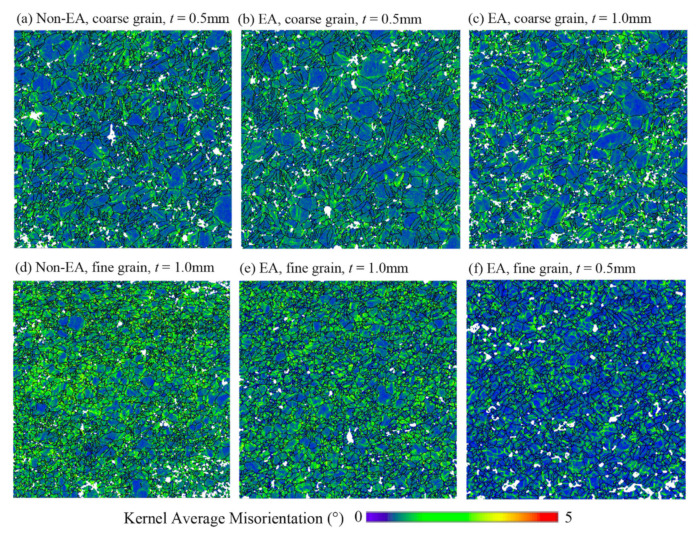
KAM maps of the samples strained to 10% tensile deformation under various deformation conditions. (**a**) Non-EA, coarse grain, *t* = 0.5 mm; (**b**) EA, coarse grain, *t* = 1.0 mm; (**c**) EA, coarse grain, *t* = 0.5 mm; (**d**) non-EA, fine grain, *t* = 1.0 mm; (**e**) EA, fine grain, *t* = 1.0 mm; (**f**) EA, fine grain, *t* = 0.5 mm.

**Figure 6 materials-15-01339-f006:**
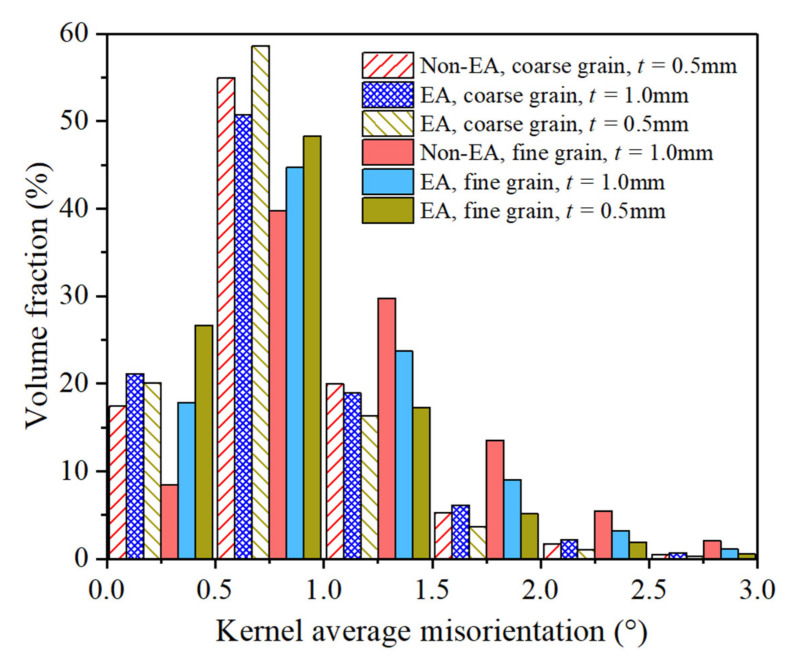
KAM histogram of the samples strained to 10% tension deformation.

**Figure 7 materials-15-01339-f007:**
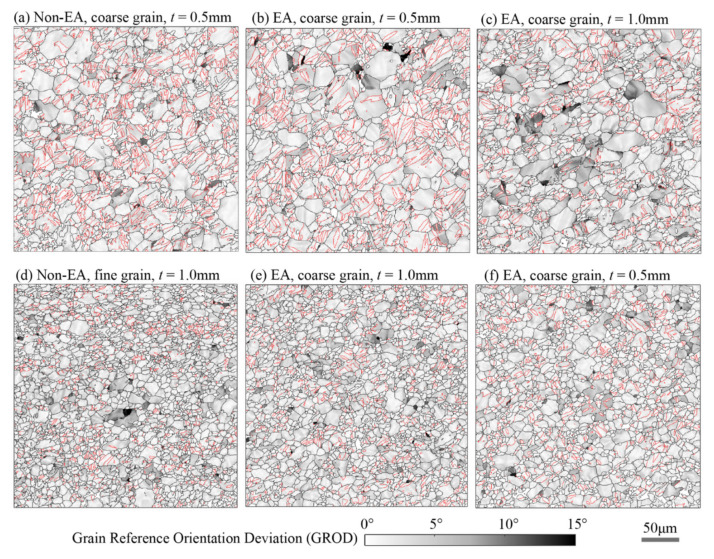
Grain reference orientation deviation (GROD) overlaid with grain boundaries (black lines) and extension twin boundaries (red lines) of the samples strained to 10% tensile deformation.

**Figure 8 materials-15-01339-f008:**
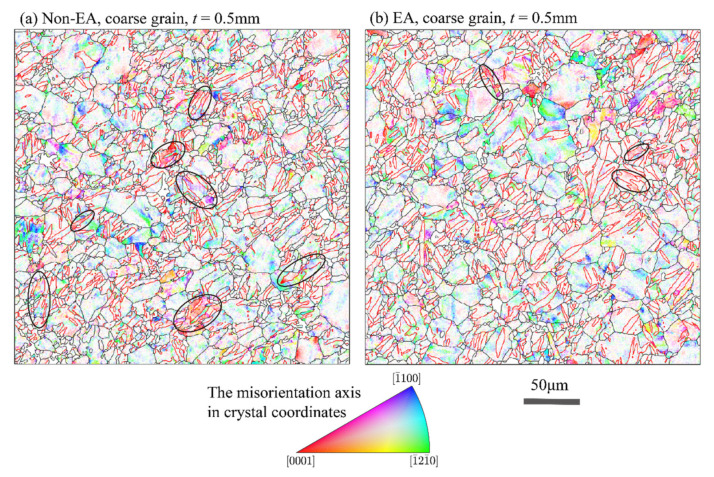
The grain reference orientation deviation (GROD) axes in crystal reference frame overlaid with grain boundaries (black lines) and extension twin boundaries (red lines) of the non-EA and EA samples strained to 10% tensile deformation. Misorientation axis vectors were colored with the hexagonal symmetry IPF, showing the axis as a direction in the HCP unit cell.

**Figure 9 materials-15-01339-f009:**
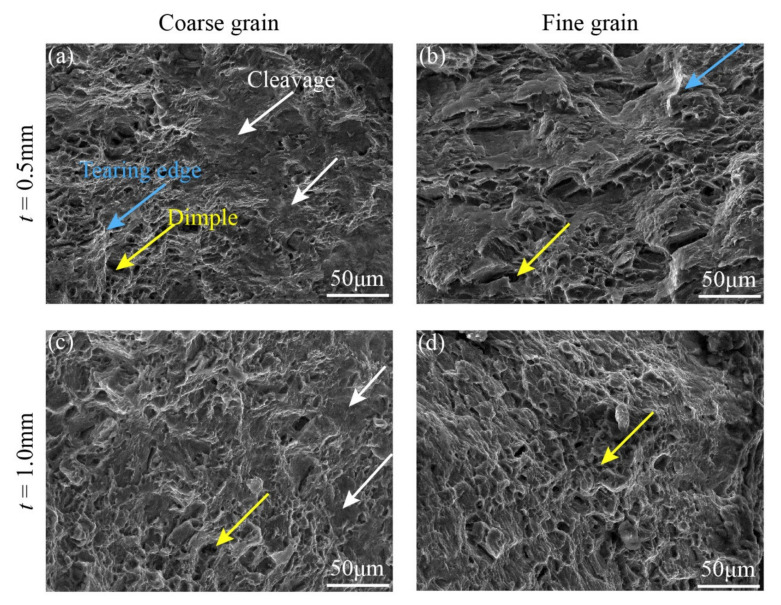
Fractography of the non-EA tensile samples under various deformation conditions of (**a**) coarse grain, *t* = 0.5 mm; (**b**) fine grain, *t* = 0.5 mm; (**c**) coarse grain, *t* = 1.0 mm; and (**d**) fine grain, *t* = 1.0 mm.

**Figure 10 materials-15-01339-f010:**
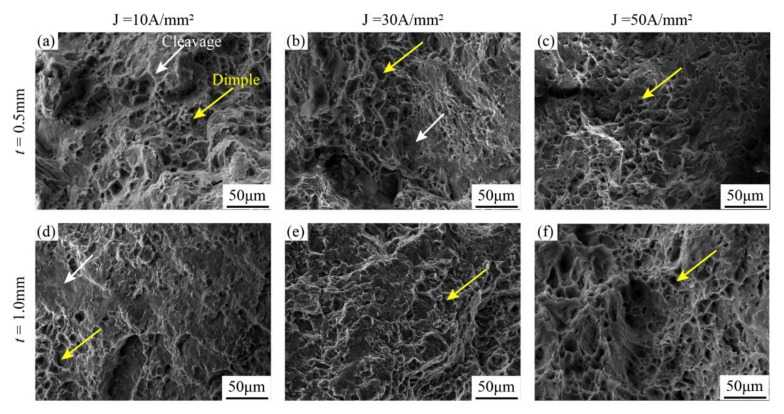
Fractography of the EA tensile samples under various deformation conditions of (**a**) coarse grain, *t* = 0.5 mm, J = 10 A/mm^2^; (**b**) coarse grain, *t* = 0.5 mm, J = 30 A/mm^2^; (**c**) coarse grain, *t* = 0.5 mm, J = 50 A/mm^2^; (**d**) fine grain, *t* = 1.0 mm, J = 10 A/mm^2^; (**e**) fine grain, *t* = 1.0 mm, J = 30 A/mm^2^; and (**f**) fine grain, *t* = 1.0 mm, J = 50 A/mm^2^.

## Data Availability

The data presented in this study are available on request from the corresponding author.

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
