# Peer review of "Electroplastic Effects on the Mechanical Responses and Deformation Mechanisms of AZ31 Mg Foils"

_materials, 2022, doi:10.3390/ma15041339_

Round 1

Reviewer 1 Report

Dear Authors,

1) Please revise minor English spell in the text such as ..."dislocation 'ship' and deformation twinning..." and  Stolyarov et al [8]'.' and "Consequently, EA metal forming operations have been receiving 'been receiving' growing interest as an important solution to form..."

2) The authors are invited to add the grain size map distribution of the initial samples (coarsed and fined grains).

3) English check "The difference in UTS of the coarse grain specimens due to sample thickness might (be) because of the different mechanical properties between the surface grains and the interior grains [26]."

4) To better understand the discussion of Figure 4, the authors are invited to add the IQ map with TB-GB which can be obtained from the samples strained to 10% tensile deformation 

5) Also, the authors are invited to add a twin boundary misorientation map to gain insigth in both contraction and extension twins which are not visible in the OIMs of Figure 4. 

6) Conclusions should consider the comments 4 and 5.

Best regards,

Reviewer # 1

Reviewer 2 Report

The title of the paper does not reflect the content of the work. The fact is that the electroplastic effect is understood as a decrease in flow stress under the action of electric current pulses with a supercritical amplitude. The temperature increase is slight in this case. Heating to the same value does not increase plasticity. This means that the electroplastic effect cannot be explained by thermal heating. This is the mystery of the electroplastic effect. The mechanism of action of current pulses is still unclear. Athermal mechanisms of deformation are being discussed. In the work, however, consideration is given to the action of direct electric current, which is reduced only to heating of the specimen.

Therefore, the title of the paper should be changed to reflect the content of the work.

Reviewer 3 Report

This manuscript deals with the electroplasticity effects on the mechanical responses and deformation mechanisms of AZ31 Mg foils by experimental study. The topic and the obtained results are very interesting, which advances the understanding of the elastoplasticity effects in the micro-scale forming of metallic materials. The reviewer is pleased to recommend this manuscript for publication in Materials after several minor revisions.

The detailed comments are stated as follows:

  1. Figure 3, the true stress shows a decrease at the end of the curve. The review guesses that this may be caused by the localized necking, not a ‘real decrease’. If so, the true stress after necking could not be directly calculated by using the engineering stress-strain curve. Therefore, two ways could be used to revise the; one is to use the engineering stress-strain curves, another is to keep using true strain-stress curves but have to remove the data after the necking point. Just for your information.
  2. Please kindly check the format of some units in the manuscript. For example, “0.3mm/s” should be changed to “0.3 mm/s” (a space should be added between the value and the unit; “70MPa” should be changed to “70 MPa”.
  3. If possible, please kindly provide the volume of grain size in Figure 2.
  4. It would be nice if an outlook could be provided in the conclusion. For example, how these interesting findings obtained in this research contribute to the manufacturing of micro-scaled products.

Thanks!

Round 2

Reviewer 1 Report

Dear Authors, 

The manuscript has been carefully revised by adding complementary results and the corresponding discussions. Moreover, the authors suggested an exciting perspective for ongoing research. Thus, the revised manuscript can be considered for publication in the Materials journal.

Yours sincerely,

Reviewer